# Melatonin Mitigates iNOS-Related Effects of HEMA and Camphorquinone in Human Dental Pulp Cells: Relevance for Postoperative Sensitivity Mechanism in Type 2 Diabetes

**DOI:** 10.3390/ijms24032562

**Published:** 2023-01-29

**Authors:** Jugoslav Ilić, Aleksandra Milosavljević, Miloš Lazarević, Maja Milošević Marković, Jelena Milašin, Milan Vučetić, Akhilanand Chaurasia, Vesna Miletić, Jelena Roganović

**Affiliations:** 1Department of Restorative Odontology and Endodontics, School of Dental Medicine, University of Belgrade, 11000 Belgrade, Serbia; 2Department of Pharmacology in Dentistry, School of Dental Medicine, University of Belgrade, 11000 Belgrade, Serbia; 3Department of Human Genetics, School of Dental Medicine, University of Belgrade, 11000 Belgrade, Serbia; 4Department of Public Health, School of Dental Medicine, University of Belgrade, 11000 Belgrade, Serbia; 5Department of Oral Surgery, School of Dental Medicine, University of Belgrade, 11000 Belgrade, Serbia; 6Department of Oral Medicine and Radiology, Faculty of Dental Sciences, King George’s Medical University, Lucknow 226003, India; 7Sydney Dental School, Faculty of Medicine and Health, University of Sydney, Sydney 2145, Australia

**Keywords:** apoptosis, camphorquinone, composite resin, dental pulp, HEMA, melatonin, postoperative sensitivity, type 2 diabetes

## Abstract

High elution and diffusion of 2-hydroxylethyl methacrylate (HEMA) and camphorquinone (CQ) through dentinal tubules may induce pulp injury and postoperative sensitivity. We aimed to investigate the melatonin protective effect in HEMA- and CQ-treated human dental pulp cells (hDPCs) as well as its relevance in a mechanism for postoperative sensitivity in diabetic patients. hDPCs were exposed to HEMA (5 mM) and/or CQ (1 mM) in the absence and presence of melatonin (MEL) (0.1 mM and 1 mM). Heme oxygenase-1 (HMOX1), NADPH oxidase-4 (NOX4), BCL-2-associated X-protein (BAX), B-cell lymphoma-2 (BCL-2) and caspase-3 (CASP3) gene expression levels, and superoxide dismutase (SOD) activity were measured in hDPCs while inducible nitric oxide synthase (iNOS) and melatonin protein expression were measured in human dental pulp as well, by RT-PCR, by ELISA, and spectrophotometrically. Bioinformatic analyses were performed by using the ShinyGO (v.0.75) application. Type 2 diabetic patients showed a higher incidence of postoperative sensitivity and lower melatonin and higher iNOS content in dental pulp tissue compared with non-diabetic patients. Melatonin, when co-added in hDPC culture, reverses HEMA and CQ cytotoxic effects via anti-apoptotic and anti-inflammatory/antioxidant iNOS-related effects. Enrichment analyses showed that genes/proteins, altered by HEMA and CQ and normalized by melatonin, are the most prominently overrepresented in type 2 diabetes mellitus pathways and that they share subcellular localization in different oligomeric protein complexes consisting of anti- and pro-apoptotic regulators. This is the first evidence of the ability of melatonin to counteract iNOS-mediated inflammatory and stress effects in HEMA- and CQ-treated hDPCs, which could be of significance for the modulation of presently observed immediate postoperative sensitivity after composite restoration in type 2 diabetic patients.

## 1. Introduction

It is well known that various components, monomers, initiators, inhibitors, and additives may elute from resin-based materials due to incomplete polymerization or biodegradation [1]. More recent work has shown higher elution and diffusion through dentinal tubules of eluates of lower molecular weight, such as 2-hydroxylethyl methacrylate (HEMA) and camphorquinone (CQ), compared with dimethacrylate monomers of higher molecular weight [2]. Clinically, this may result in postoperative sensitivity, pulpitis, and potentially even pulp necrosis [3]. It has been shown that both HEMA and CQ may cause cytotoxicity, cell cycle arrest, and reactive oxygen species (ROS)-mediated genotoxicity in human oral cells [4,5,6]. Despite the use of HEMA and CQ as a reactant mixture during adhesive application, the literature lacks data on their concomitant effect on human dental pulp cells (hDCPs). Major sources of ROS in many cell types are an NADPH oxidase family of enzymes, while nitric oxide modulates NADPH oxidase activity via *heme oxygenase-1 (HMOX1*), [7]. Nitric oxide plays an essential role in pain responses to injury and inflammation in dental pulp [8] and contributes to hyperalgesia (increased sensitivity to painful stimuli) via peroxynitrite formation [9]. A major source of NO in inflammation/injury is inducible nitric oxide synthase (iNOS), shown to have an important peripheral role in the development of inflammatory and neuropathic pain. Moreover, iNOS plays a key role in peroxynitrite injury to peripheral nerves as well as in functional and structural changes of nerves and pain caused by diabetes, known as diabetic peripheral neuropathy [10]. At the same time, epidemiological data show that type 2 diabetes is associated with greater orofacial pain [11]. Nevertheless, there are no data of iNOS-related diabetic pain alterations influencing postoperative sensitivity after composite restoration placement.

Both in vitro and in vivo studies suggested that melatonin has scavenging activity on highly toxic hydroxyl and oxygen radicals [12], inhibits inflammation induced by iNOS, and attenuates hyperalgesia [13]. Furthermore, by reducing the number of NOS-positive cells, melatonin induces substantial attenuation of trigeminovascular nociception [14]. Thus, we hypothesized that the iNOS level is increased in diabetic pulp with composite restoration and that melatonin could beneficially interfere with the mechanism of immediate postoperative sensitivity after composite restoration, induced by HEMA and CQ, by attenuating the iNOS-mediated inflammatory and stress effects of HEMA and CQ in hDPCs, which could be important for diabetic patients especially. Therefore, we aimed to investigate: (1) iNOS/MEL levels in the human diabetic pulp of restored teeth and the incidence of postoperative sensitivity after composite restoration in type 2 diabetic patients; and (2) melatonin effects on iNOS-related pathways in HEMA- and CQ-treated hDPCs. 

## 2. Results

### 2.1. Melatonin and iNOS Content in Dental Pulp Tissue from Non-Diabetic and Type 2 Diabetic Patients and Incidence of Postoperative Sensitivity after Placement of Direct Composite Restorations in Type 2 Diabetic Patients

A comparison of iNOS and MEL protein expression in dental pulp tissue of teeth with composite restorations from non-diabetic (n = 7) and diabetic (n = 6) participants revealed significantly lower MEL and significantly higher iNOS content in diabetic vs. non-diabetic dental pulp (Figure 1).

The comparison between age- and treatment-matched groups showed a higher incidence of immediate postoperative sensitivity (24–48 h) in type 2 diabetic compared with non-diabetic patients after restoration with composite filling using the HEMA-containing adhesive OptiBond Solo Plus (Kerr, Brea, CA, USA) (Table 1).

### 2.2. Melatonin Effects on Viability of hDPCs Treated with HEMA and/or CQ

MTT assay results showed no significant changes in hDPC viability when low (MEL1 = 0.1 mM) and high (MEL2 = 1 mM) pharmacological concentrations of melatonin were applied (Figure 2). DMSO and ethanol, as solvents, were also without effect on hDPC viability. After exposure to HEMA, CQ, and HEMA + CQ (24 h), a significant reduction in hDPC viability was observed compared with control: 86.80 ± 4.50%, 87.58 ± 6.37%, and 74.38 ± 6.78%, respectively, indicating that injurious effect on hDPCs was more pronounced when HEMA and CQ were applied concomitantly. The addition of MEL2 potentiated the effects of HEMA and CQ. When MEL1 was added, the number of viable cells was returned to normal in the CQ-supplemented culture (104.6 ± 7.99%), and cell viability was significantly ameliorated in HEMA- and HEMA + CQ-supplemented culture, up to 93.49 ± 4.86% and 88.87 ± 3.45%, respectively (Figure 2).

### 2.3. Melatonin Effects on BAX, BCL-2, and CASP3Gene Expression Levels in hDPCs Treated with HEMA and/or CQ

While HEMA decreases *BCL-2*, the addition of MEL2 is effective in normalizing *BCL-2* expression levels after HEMA treatment (Figure 3B). Concomitant use of HEMA and CQ significantly increases *caspase-3*; MEL1, per se, decreases its level and normalizes its increased levels induced by concomitant HEMA and CQ treatment (Figure 3C). There are no significant changes in *BAX* gene expression levels either by HEMA and/or CQ, or by any applied melatonin concentration (Figure 3A).

### 2.4. Melatonin Effects on HMOX1 and NOX4 Gene Expression Levels in hDPCs Treated with HEMA and/or CQ

RT-qPCR analysis showed that MEL1 (0.1 mM) significantly reduced *HMOX1* mRNA levels in hDPCs, while treatment with MEL2 (1 mM) caused up-regulation of *HMOX1* mRNA and down-regulation of *NOX4* mRNA levels when compared with control. HEMA and CQ significantly up-regulated *HMOX1* mRNA levels in hDPCs, being elevated up to 5.5-fold after treatments with HEMA and HEMA + CQ, and up to 2.5-fold after treatment with CQ (Figure 4A), while they significantly down-regulated *NOX4* mRNA levels, up to 6.4, 2.5, and 9.6-fold after treatment with HEMA, CQ, and HEMA + CQ, respectively (Figure 4B). The addition of MEL1 and MEL2 mitigated a HEMA- and/or CQ-induced increase in *HMOX1* mRNA, and MEL1 reversed the effects of HEMA and/or CQ on *NOX4* mRNA levels, while MEL2 potentiated the effects of CQ and HEMA + CQ on *NOX4* mRNA levels in hDPCs (Figure 4).

### 2.5. Melatonin Effects on SOD Activity and iNOS Protein Levels in hDPCs Treated with HEMA and/or CQ

Treatments with MEL1 (0.1 mM) and MEL2 (1 mM) were without significant changes in SOD activity compared with control. After hDPC incubation with HEMA, CQ, and HEMA + CQ (24 h), SOD activity was significantly increased, being elevated up to 1.81, 1.77, and 1.88-fold, respectively, when compared with control. The addition of both pharmacological concentrations of melatonin to HEMA and/or CQ significantly reduced HEMA- and/or CQ-induced up-regulation of SOD activity (Figure 5A). Results of ELISA analysis showed significantly increased iNOS protein levels after hDPC treatment with MEL2, HEMA, CQ, and HEMA + CQ, being elevated up to 2.0, 1.81, 1.52, and 1.7-fold, respectively. The addition of MEL1 reversed the effects of HEMA and HEMA + CQ on iNOS protein levels in hDPCs, while the addition of MEL2 caused more pronounced increases in iNOS protein levels (Figure 5B). There was a significant positive correlation between *HMOX1* mRNA expression and iNOS protein expression found in hDPCs treated with HEMA (r = 0.99, *p* ˂ 0.05) and HEMA + CQ (r = 0.98, *p* ˂ 0.05) in the presence of MEL1.

### 2.6. Enrichment Analyses and Gene Protein–Protein Interaction Network

There were 15 interactions found between investigated genes/proteins: HMOX1, NOS2 (iNOS), SOD1, SOD2, SOD3, and CASP3 (Figure 6B). GO Enrichment analysis on genes involved in HEMA and CQ effects in hDPCs showed that, in terms of cellular components, most genes are related to the perinuclear region of cytoplasm, and most of the proteins shared subcellular compartments consisting of anti- and pro-apoptotic regulators (Figure 6A), while all of these genes are overrepresented in the pathways of diabetes mellitus type 2 (Figure 6C).

## 3. Discussion

This is the first evidence of the ability of melatonin to counteract iNOS-mediated inflammatory and stress effects in HEMA- and CQ-treated hDPCs, which could be a mechanism of significance for the modulation of presently observed immediate postoperative sensitivity after composite restoration in type 2 diabetic patients (Figure 7). Namely, we identified that type 2 diabetic patients have increased dental pulp levels of iNOS, an established nociceptive mediator of pulp inflammatory pain, and that the dental composite components, HEMA and CQ, especially if concomitantly present, induced cytotoxic and inflammatory/stress responses via iNOS pathways. Bearing in mind the study of Putzey et al. [2] which showed that HEMA, compared with all the other eluted monomers, was released in the highest quantities, followed by CQ, and that it had the potential to completely migrate through a 300 μm thick dentin disk (mimicking the situation of deep carious lesions which are treated), it could be presumed that the activation of iNOS by HEMA and CQ could aggravate iNOS content already deteriorated in diabetic dental pulp and affect iNOS-mediated nociception in diabetic patients.

Present results show that melatonin, when co-added in hDPC culture, reverses HEMA and CQ cytotoxic effects via anti-apoptotic and anti-inflammatory/antioxidant iNOS-related effects. Likewise, we previously showed that melatonin normalizes iNOS levels in hyperglycemic hDPCs [15]. Since it has been revealed that melatonin exerts a trigeminal antinociceptive role by decreasing the NO pathway [14], present findings may suggest the potential of locally applied melatonin in the attenuation of postoperative sensitivity in diabetic patients. Further in vivo studies should be employed to investigate this matter. 

The effects of two pharmacological concentrations of MEL were investigated in hDPCs. Both concentrations were not cytotoxic, per se, for hDPCs; however, while the co-incubation with low pharmacological MEL concentration was able to mitigate the cytotoxic effects of CQ and HEMA in hDPCs, the co-incubation with high pharmacological MEL concentration potentiated a HEMA- and CQ-induced decrease in cell viability. The administration of low melatonin induced down-regulation of *caspase-3* and *HMOX1* gene expression levels; it also normalized both *HMOX1* and *NOX4* genes as well as iNOS protein levels and SOD activity in hDPCs treated with HEMA and/or CQ. At the same time, high MEL administration was associated with *HMOX1* and iNOS up-regulation and *NOX4* down-regulation. Heme oxygenase-1 (HO-1 encoded by the HMOX1 gene) is induced rapidly after oxidative stress, and although a low expression level of HO-1 is described as a protective mechanism against oxidative stress [16], an excess expression of this enzyme was found to be associated with increased oxygen cytotoxicity and apoptosis [17,18]. HEMA- and CQ-induced up-regulation of the *HMOX1* gene correlates with iNOS protein expression, suggesting that activation of the iNOS/*HMOX1* pathway may contribute to cytotoxicity, as *HMOX1* was identified as a critical instigator of NO-derived oxidative stress and promotion of peroxynitrite production, and as a known cytotoxic mediator [19]. Low MEL normalized SOD activity in HEMA- and CQ.-treated cells, suggesting a reduction in superoxide ion generation, which is the main stimulus for SOD activity. Unlike the other NOX isoforms, *NOX4* is constitutively active, and by producing hydrogen peroxide instead of O^2−^, it does not interact with NO to produce peroxynitrite, thus being protective against cell death [20,21,22]. Bearing this in mind, it seems that besides up-regulation of *HMOX1*, iNOS, and SOD activity, down-regulation of *NOX4* is another contributing factor to HEMA- and CQ-mediated stress effects in hDPCs.

By employment of bioinformatic analyses, we found that genes/proteins, altered by HEMA and CQ, and normalized by melatonin, are the most prominently overrepresented in type 2 diabetes mellitus pathways, and that they share subcellular localization in different oligomeric protein complexes consisting of anti- and pro-apoptotic regulators, pointing at their predicted involvement in cell differentiation/death control. These results suggest that besides being involved in the modulation of diabetes-induced pulp pain, melatonin may express overall dental pulp protective effects in type 2 diabetes mellitus, as we suggested previously [15].

This is the first report of the more frequent postoperative sensitivity in type 2 diabetic vs. nondiabetic patients, and it suggests that melatonin may act beneficially due to its regulation of iNOS-related pathways in hDPCs. Further research is certainly needed to confirm melatonin effects in vivo, but this study traces the way for the future use of melatonin in composite filling restoration in order to alleviate the effects of HEMA and CQ on pulp cells, especially in diabetes.

## 4. Materials and Methods

The study protocol was approved by the Ethics Committee of the School of Dental Medicine (approval number 36/2).

### 4.1. Human Dental Pulp Tissue Collection

The non-diabetic (n = 7) and diabetic (n = 6) participants (55–65 years, both genders) were recruited at the Clinic for Restorative Odontology and Endodontics, School of Dental Medicine, University of Belgrade. Due to preprosthetic endodontic management, vital pulp extirpation was performed in teeth with composite restorations without signs of pulpal or periodontal tissue pathology, as confirmed by electric pulp testing and probing measurements. all participants were without systemic diseases, while the type 2 diabetic participants had at least three years of well-controlled type 2 DM (HbA1c < 7%).

### 4.2. Cell Culture

hDPCs were isolated from coronal and radicular dental pulp tissue explants of intact wisdom teeth with completed root development indicated for extraction due to orthodontic reasons. Wisdom teeth were collected at the Clinic for Oral Surgery, School of Dental Medicine, from two healthy patients aged 20–25 after obtaining written informed consent. The hDPCs were isolated using an outgrowth method [23]; in short, dental pulp tissue was dissected into small pieces and transferred to the cell culture flask with subsequent addition of serum-containing cell growth media (low-glucose Dulbecco’s modified Eagle’s medium, 10% fetal bovine serum, 100 units/mL of penicillin-G, and 100 μg/mL of streptomycin). All chemicals used in the research were purchased from Sigma-Aldrich (St. Louis, MO, USA). Cell cultures were maintained in a humidified 5% CO_2_ incubator with 95% air and at 37 °C. At the confluence, approximately 80% cells were subcultured using TrypLE Express (Gibco Life Technologies Inc., Grand Island, NY, USA) for cell detachment from the bottom of the flask. All experiments were performed with cells between the 3rd and 5th passages.

### 4.3. Retrospective Chart Review for Detection of Postoperative Hypersensitivity

We reviewed charts from the electronic database of the Clinic for Restorative Odontology and Endodontics, from May–November 2022, in order to collect details of postoperative sensitivity in type 2 diabetic patients after restorative treatment of primary caries on one proximal surface of a molar tooth. A comparison was made with an age- and treatment-matched group of non-diabetic patients. The inclusion criteria were: established diagnosis of dentinal caries with no pulpal or periapical involvement (confirmed radiographically and by electric pulp testing); no records of cervical lesions and periodontal disease on restored teeth; and a one-appointment restoration of a proximal lesion using composite filling with no base or liner beneath. Included patients were not on anti-inflammatory or antibiotic therapy, while those with type 2 diabetes had well-controlled type 2 DM (HbA1c < 7%) for at least three years. All observed teeth were restored with Herculite XRV (Kerr, Brea, CA, USA) using HEMA-containing adhesive OptiBond Solo Plus (Kerr, Brea, CA, USA). Commencement of immediate sensitivity was considered if there was a record of pain in the 24 to 48 h period after the filling placement.

### 4.4. MTT Assay

In order to examine the viability of the hDPCs after exposure to HEMA (5 mM) and/or CQ (1 mM) and in the presence of pharmacological concentrations of melatonin (MEL1 = 0.1 mM and MEL2 = 1 mM), an MTT assay was performed because only viable and metabolically active cells can reduce MTT tetrazolium salt [3-(4,5-Dimethylthiazol-2-yl)-2,5-diphenyltetrazolium bromide] into insoluble purple formazan crystals [24]. The hDPCs were seeded into 96-well plates (104 cells/well in 100 µL of cell culture media). After 24 h of incubation, the cell culture media were removed and the cells were treated for next 24 h with new cell culture media containing solvents (0.4% dimethylsulfoxide, DMSO, and 0.25% ethanol) or aforementioned concentrations of the tested chemicals: MEL1, MEL2, HEMA, CQ, HEMA + CQ, MEL1 + HEMA, MEL1 + CQ, MEL1 + HEMA + CQ, MEL2 + HEMA, MEL2 + CQ, and MEL2 + HEMA + CQ, including a blank group (wells without seeded cells). DMSO was used for CQ dissolving, and 96% ethanol was used for melatonin dissolving. After the indicated time of incubation (24 h), the cell culture media were removed, and all wells were washed once with 150 µL of Dulbecco’s Phosphate Buffered Saline with subsequent addition of 100 µL cell culture media containing the MTT at a concentration of 0.5 mg/mL, and then incubated for three hours in an incubator protected from the light. Finally, after aspiration of the MTT solution, 100 µL of DMSO was added to dissolve formazan crystals with subsequent plate shaking at 37°C for 10 min at 250 rpm, and then optical density (OD) was read at 540 nm by a plate-reading spectrophotometer (RT-2100C, Transwin Medical Equipment Co., Zhejiang, China). To estimate cell viability, the following formula was used: Cell viability (%) = (mean OD of treated cells—mean OD of blank/mean OD of solvent-treated cells—mean OD of blank) × 100. The MTT assay was performed on the hDPC cultures isolated from two patients and repeated two times for each.

### 4.5. Real-Time Quantitative Polymerase Chain Reaction (RT-qPCR)

The hDPCs were seeded in 12-well plates and, after 24 h of incubation, were exposed to the same treatment regime as for the MTT assay: solvents (0.4% DMSO and 0.25% ethanol), MEL1, MEL2, HEMA, CQ, HEMA + CQ, MEL1 + HEMA, MEL1 + CQ, MEL1 + HEMA + CQ, MEL2 + HEMA, MEL2 + CQ, and MEL2 + HEMA + CQ. Solvent-treated hDPCs were used as control. After incubation time (24 h), total RNA was extracted from cells with TRIzol Reagent (Invitrogen, Thermo Fisher Scientific, Waltham, MA, USA), and first-strand cDNA was synthesized from 2 μg of total RNA using Oligo (dT) primer and RevertAid Reverse Transcriptase (RT) (Thermo Fisher Scientific, Waltham, MA, USA). For PCR analysis, cDNA was amplified by Taq DNA polymerase. Subsequent RT-qPCR analysis was performed on the Line Gene-K Fluorescence Real-time PCR Detection System (Bioer, China) using the Maxima™ SYBR Green/ROX qPCR Master Mix (Thermo Fisher Scientific, Waltham, MA, USA). The expression of target genes: NOX4—NADPH oxidase-4 (forward: 5′-CTCAGCGGAATCAATCAGCTGTG-3′; reverse: 5′ AGAGGAACACGACAATCAGCCTTAG-3′), HMOX1—heme oxygenase-1 (forward: 5′-TTTGAGGAGTTGCAGGAGC-3′; reverse: 5′-AGGACCCATCGGAGAAGC-3′), BAX—BCL-2-associated X-protein (forward: 5′-ATGTTTTCTGACGGCAACTTC-3′; reverse: 5′-AGTCCAATGTCCAGCCCAT-3′), BCL-2—B-cell lymphoma-2 (forward: 5′-ATGTGTGTGGAGAGCGTCAACC-3′; reverse: 5′-TGAGCAGAGTCTTCAGAGACAGCC-3′), and CASP3—caspase-3 (forward: 5′-TGTTTGTGTGCTTCTGAGCC-3′; reverse: 5′-CACGCCATGTCATCATCAAC-3′). The housekeeping gene GAPDH—glyceraldehyde 3-phosphate dehydrogenase (forward: 5′-TCATGACCACAGTCCATGCCATCA-3′; reverse: 5′-CCCTGTTGCTGTAGCCAAATTCGT-3′), used as an endogenous standard, was analyzed under the same conditions. The comparative ΔΔCt method was used for the relative quantification of gene expression as described by Livak and Schmittgen [25].

### 4.6. Spectrophotometric Analysis

The hDPCs were seeded in 6-well plates and, at confluence of 80%, were exposed to the same treatment regime as for the MTT assay: solvents–control (0.4% DMSO and 0.25% ethanol), MEL1, MEL2, HEMA, CQ, HEMA + CQ, MEL1 + HEMA, MEL1 + CQ, MEL1 + HEMA + CQ, MEL2 + HEMA, MEL2 + CQ, and MEL2 + HEMA + CQ. After 24 h of incubation, the hDPCs were collected, with subsequent protein extraction using a freeze–thaw technique and sample centrifugation (10.000× *g*/5 min), followed by supernatant collection and total protein quantification by the BioSpecnano Micro-volume UV–Vis spectrophotometer (Shimadzu Scientific Instruments, Columbia, MD, USA). In the supernatants, superoxide dismutase (SOD) activity (%) was quantified spectrophotometrically by a SOD Determination Kit (Sigma-Aldrich, St. Louis, MO, USA) following the manufacturer’s instructions.

### 4.7. Enzyme-Linked Immunosorbent Assay (ELISA)

For iNOS quantification, the same cell supernatants were used as for the SOD activity determination. The iNOS (pg/mL) was quantified using commercial ELISA kits according to the manufacturer’s instructions [Human Inducible nitric oxide synthase (iNOS) ELISA Kit, Biopeony, Beijing, China].

### 4.8. Bioinformatic Analyses

To analyze known and predicted gene/protein interactions, human interactome data were used for the creation of a protein–protein network from the StringApp11.5 (Search Tool for the Retrieval of Interacting Genes/Proteins) database (https://string-db.org, accessed on 17 December 2022) [26] for melatonin-acting genes HMOX1, NOS2 (iNOS), SOD, and CASP3. The interaction network was composed of a set of genes/proteins (nodes) connected by colored lines which represent different functional relationships among these genes/proteins. For the network visualization, Gene Ontology (GO) [27] diseases enrichment, Kyoto Encyclopedia of Genes and Genomes (KEGG) (http://www.genome.jp/kegg/pathway.html) [28] and compartments [29] analyses, we used the ShinyGO (v.0.75) application [30] (Available online: http://bioinformatics.sdstate.edu/go/, accessed on 17 December 2022).

### 4.9. Data Analyses

Statistical analyses were performed using the GraphPad Prism 9 software package (Graph Pad Software Inc., San Diego, CA, USA). The results were presented as mean ± SD. The Mann–Whitney test was used for testing the difference between two groups (each chemical vs. control and chemical with MEL vs. chemical without MEL). Descriptive statistics and the z test were used for data on the incidence of postoperative sensitivity. A *p*-value of < 0.05 was considered statistically significant. FDR is calculated based on the nominal *p*-value from the hypergeometric test. Fold Enrichment is defined as the percentage of genes belonging to a pathway divided by the corresponding percentage in the background. Fold Enrichment indicates how genes of a certain pathway are overrepresented.

## 5. Conclusions

Concomitant use of HEMA and CQ potentiates cytotoxic effects in hDPCs as reflected by decreased viability, up-regulation of caspase-3, HMOX1, iNOS expression and SOD activity, and down-regulation of BCL-2 and NOX4. This is the first evidence of the ability of melatonin to counteract iNOS-mediated inflammatory and stress effects in HEMA- and CQ-treated hDPCs, which could be of significance for the modulation of presently observed immediate postoperative sensitivity after composite restoration in type 2 diabetic patients.

## Figures and Tables

**Figure 1 ijms-24-02562-f001:**
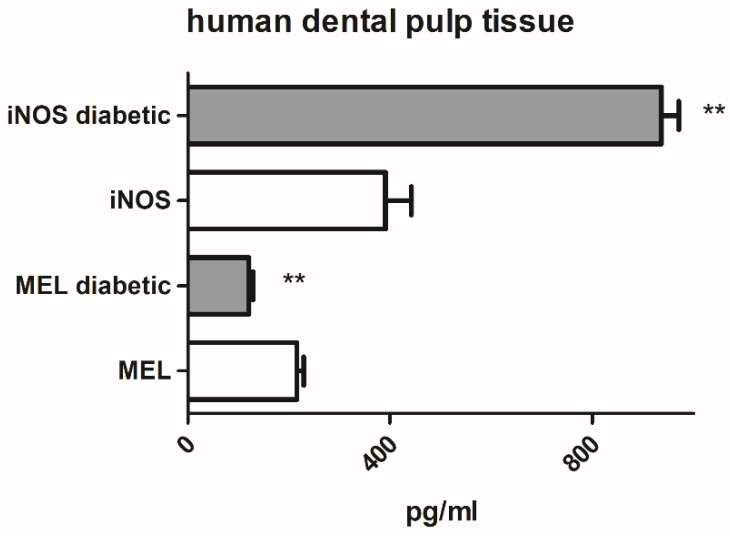
Inducible nitric oxide synthase (iNOS) and melatonin (MEL) protein expression in human dental pulp tissue from non-diabetic and type 2 diabetic patients. Each graph bar represents the mean ± SD of six to seven samples. ** *p* < 0.01.

**Figure 2 ijms-24-02562-f002:**
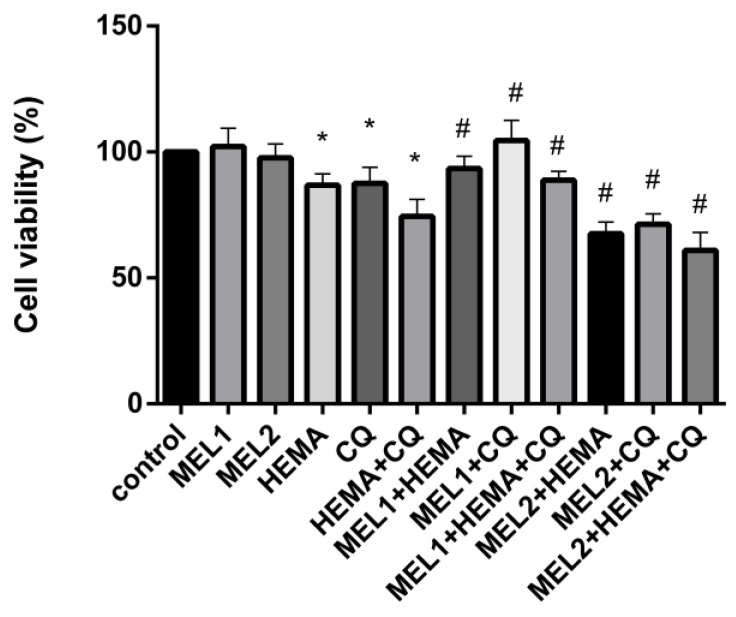
Human dental pulp cells viability (%) after treatment with HEMA and/or CQ in the presence of melatonin (24 h) assessed by MTT assay. Each graph bar represents the mean ± SD of four experiments. * *p* < 0.05 chemical vs. control; # *p* < 0.05 chemical with melatonin vs. without melatonin. HEMA—2-hydroxyethyl methacrylate (5 mM); CQ—camphorquinone (1 mM); and MEL1 (0.1 mM) and MEL2 (1 mM)—melatonin.

**Figure 3 ijms-24-02562-f003:**
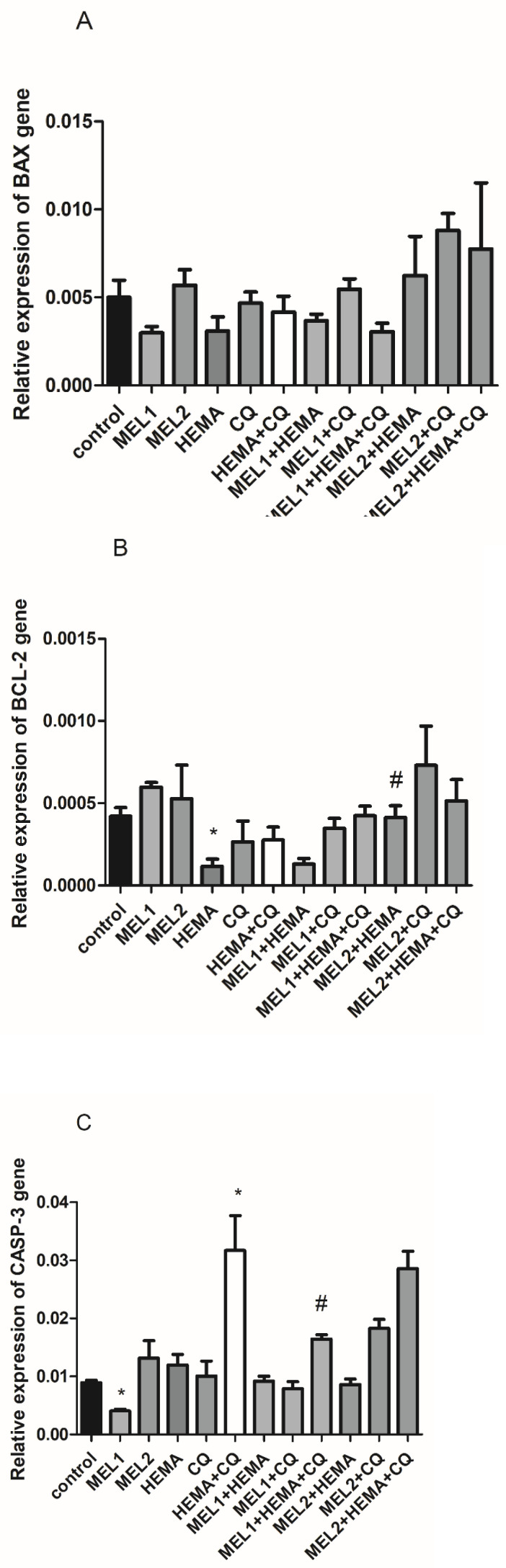
HEMA- and/or CQ-induced effects on *BAX* (**A**), *BCL-2* (**B**), and *CASP3* (**C**) gene expression levels in human dental pulp cells in the presence of melatonin (24 h). Each graph bar represents the mean ± SD of four experiments. * *p* < 0.05 chemical vs. control; # *p* < 0.05 chemical with melatonin vs. without melatonin. BAX—BCL-2-associated X-protein; BCL-2—B-cell lymphoma-2; CASP3—caspase-3; MEL1 (0.1 mM) and MEL2 (1 mM)—melatonin; HEMA—2-hydroxyethyl methacrylate (5 mM); and CQ—camphorquinone (1 mM).

**Figure 4 ijms-24-02562-f004:**
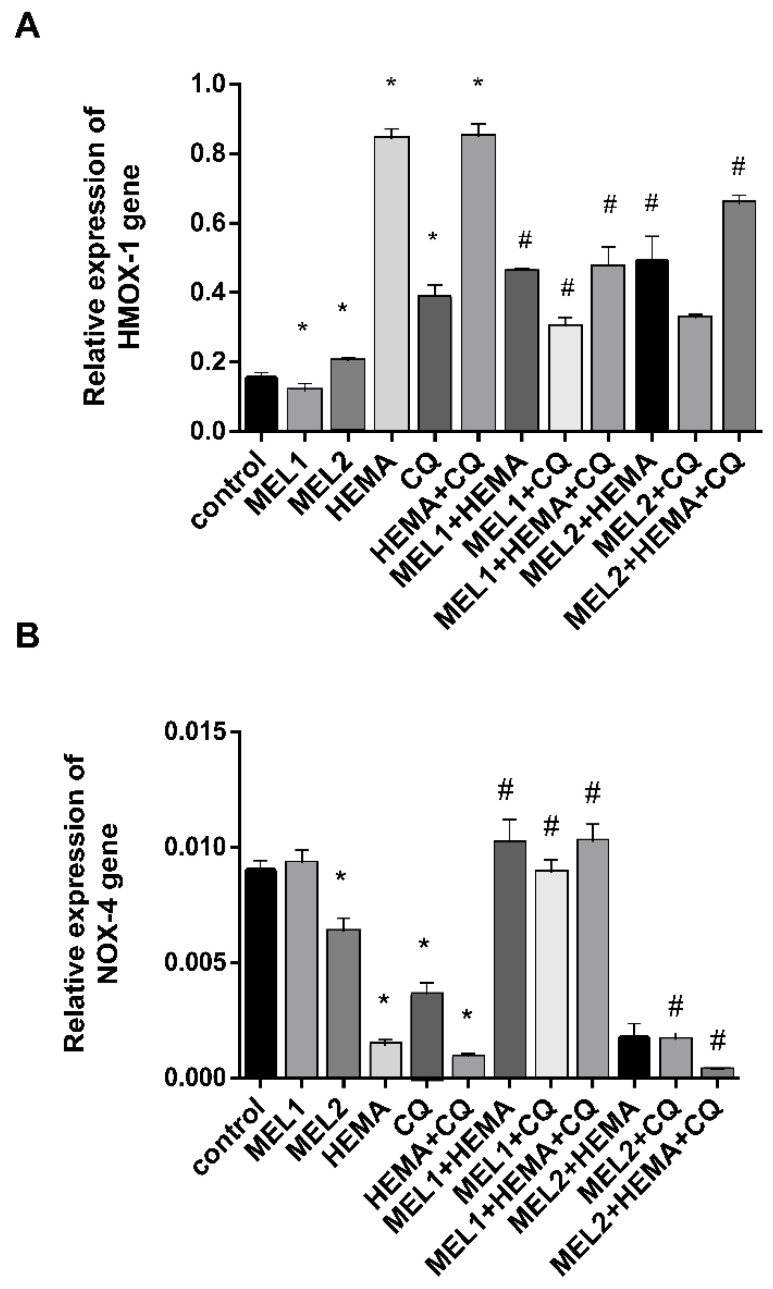
HEMA- and/or CQ-induced effects on *HMOX1* (**A**) and *NOX4* (**B**) gene expression levels in human dental pulp cells in the presence of melatonin (24 h). Each graph bar represents the mean ± SD of four experiments. * *p* < 0.05 chemical vs. control; # *p* < 0.05 chemical with melatonin vs. without melatonin. HMOX1—heme oxygenase-1 gene; NOX4—NADPH oxidase-4 gene; MEL1 (0.1 mM) and MEL2 (1 mM)—melatonin; HEMA—2-hydroxyethyl methacrylate (5 mM); and CQ—camphorquinone (1 mM).

**Figure 5 ijms-24-02562-f005:**
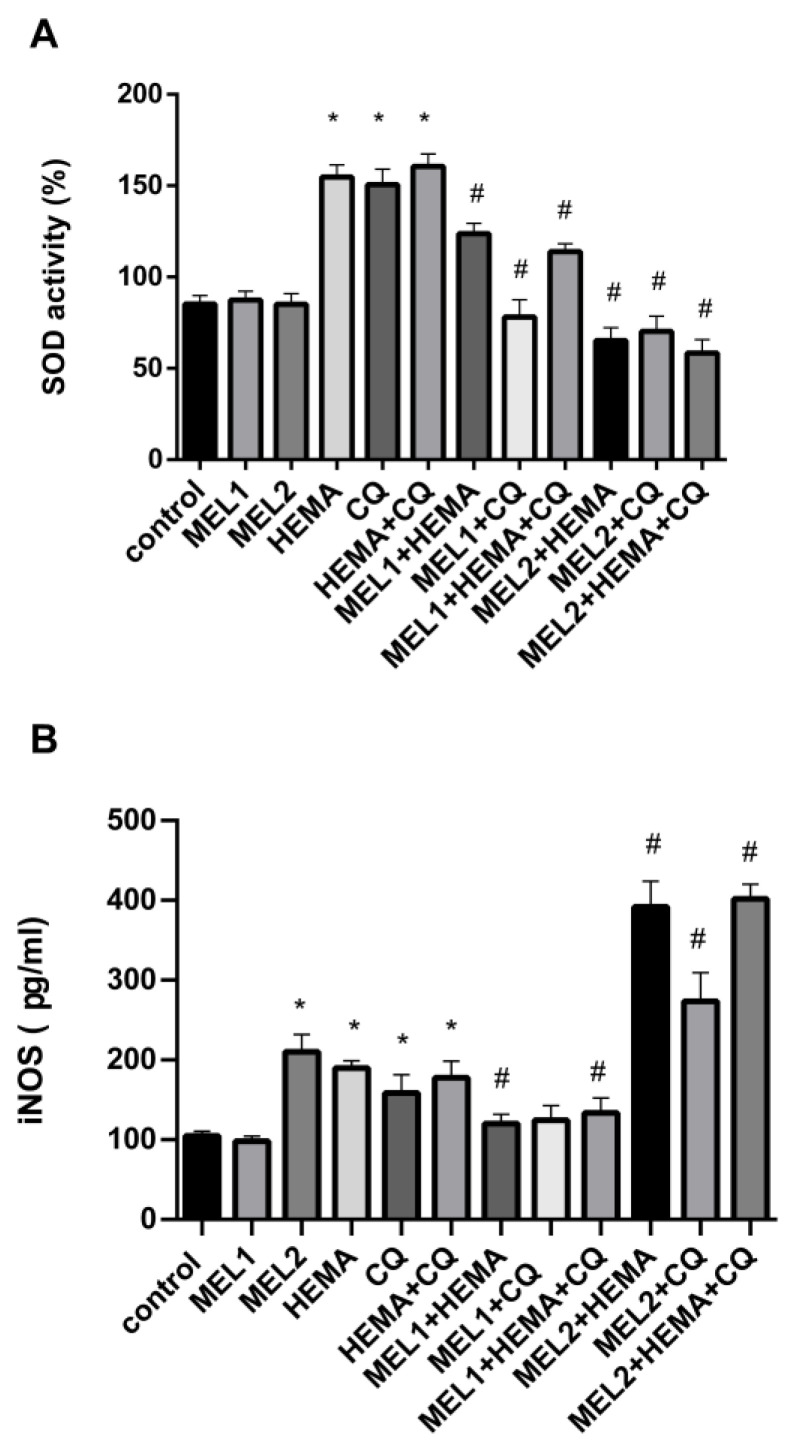
HEMA- and/or CQ-induced effects on SOD activity (**A**) and iNOS protein expression levels (**B**) in human dental pulp cells in the presence of melatonin (24 h). Each graph bar represents the mean ± SD of four experiments. * *p* < 0.05 chemical vs. control; # *p* < 0.05 chemical with melatonin vs. without melatonin. SOD—superoxide dismutase; iNOS—inducible nitric oxide synthase; MEL1 (0.1 mM) and MEL2 (1 mM)—melatonin; HEMA—2-hydroxyethyl methacrylate (5 mM); and CQ—camphorquinone (1 mM).

**Figure 6 ijms-24-02562-f006:**
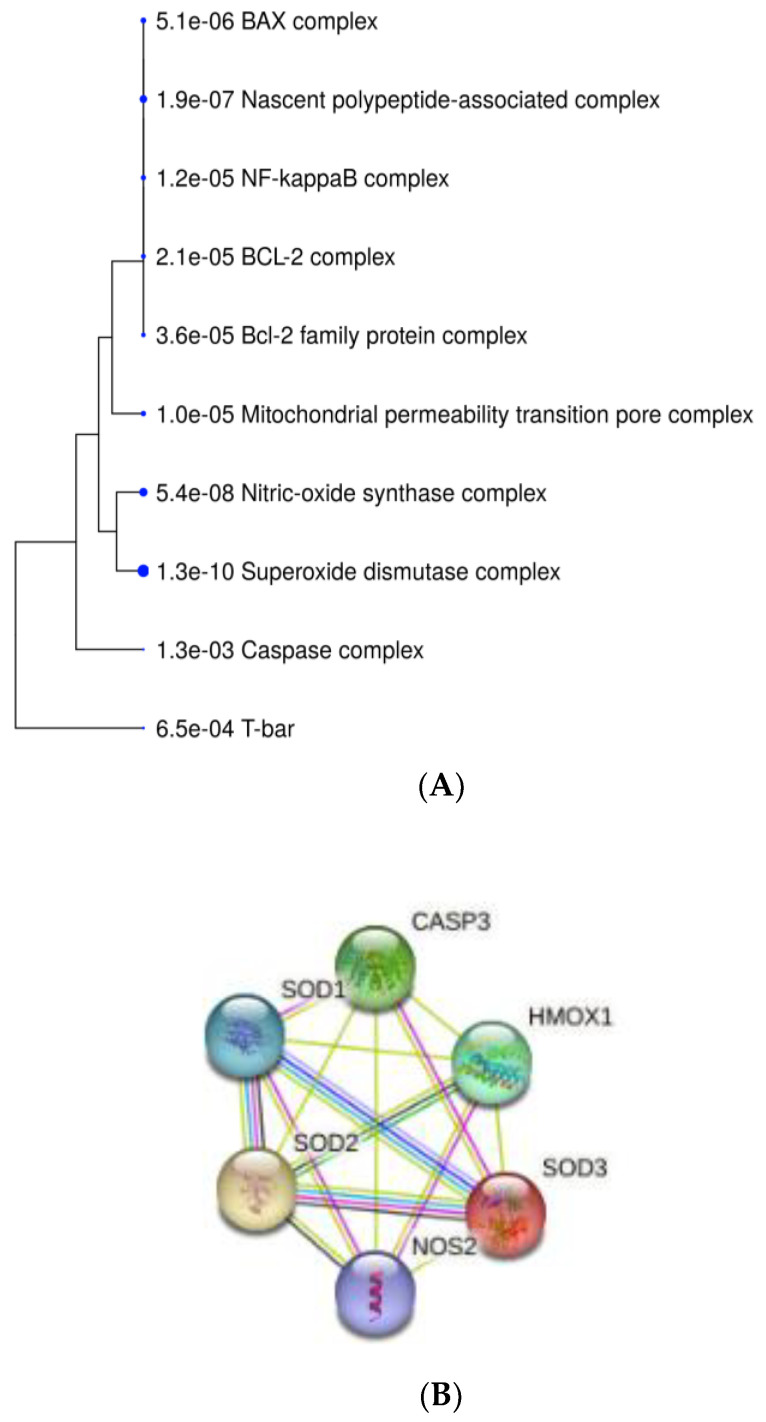
Bioinformatic analyses of genes/proteins involved in MEL effects in HEMA- and/or CQ-treated human dental pulp cells. A hierarchical clustering tree summarizing the correlation among proteins sharing subcellular localization in compartments. Compartments with many shared genes are clustered together (**A**). Predicted gene/protein interaction network of HMOX1, NOS2 (iNOS) SOD and CASP3. Nodes represent the genes/proteins connected by edges which represent functional relationships among them (**B**). GO Enrichment analysis results regarding involvement in diseases: the x-axis represents the number of genes enriched in each GO, and the y-axis represents the diseases involved (**C**). HMOX1—heme oxygenase-1; CASP3—caspase-3; NOS2 (iNOS)—inducible nitric oxide synthase; SOD-superoxide dismutase; known interactions from curated databases (blue line); known interactions experimentally determined (purple line); gene neighborhood (green line); gene co-expression (black line); text mining (yellow line); FDR is calculated based on nominal *p*-value from the hypergeometric test. Fold Enrichment is defined as the percentage of genes belonging to a pathway divided by the corresponding percentage in the background. Fold Enrichment indicates how genes of a certain pathway are overrepresented.

**Figure 7 ijms-24-02562-f007:**
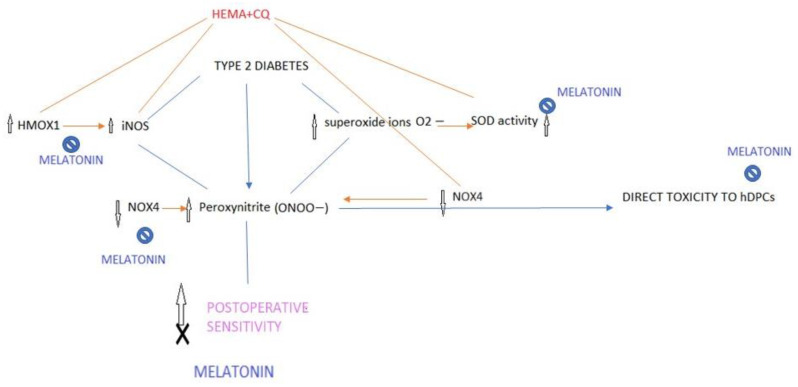
Proposed modus of melatonin interfering with the mechanism of postoperative sensitivity in type 2 diabetes. Diabetes is associated with greater orofacial pain while iNOS plays a key role in peroxynitrite injury of peripheral nerves and nerve pain caused by diabetes (peripheral neuropathy). By decreasing iNOS (HMOX1/iNOS) and superoxide ion generation (SOD activity) as well as by down-regulation of NOX4 in HEMA- and CQ-treated hDPCs, melatonin may attenuate peroxynitrite-mediated nociception and postoperative sensitivity induced by HEMA and CQ.

**Table 1 ijms-24-02562-t001:** Incidence of postoperative sensitivity after placement of composite restoration in type 2 diabetic and non-diabetic patients.

	Group	
	Non-Diabetic	Type 2 Diabetic
Number of patients Age (mean ± SEM) Number of fillings Fillings with postoperative sensitivity n (% of total)	32 55.2 ± 3.3 44 5 (11.4%)	32 58.6 ± 3.0 87 25 (28.7%) *

* *p* < 0.05 type 2 diabetic vs. non-diabetic patients.

## Data Availability

The data presented in this study are available on request from the corresponding author.

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
