# Peer review of "Melatonin Mitigates iNOS-Related Effects of HEMA and Camphorquinone in Human Dental Pulp Cells: Relevance for Postoperative Sensitivity Mechanism in Type 2 Diabetes"

_ijms, 2023, doi:10.3390/ijms24032562_

Round 1

Author Response

REVIEWER 1

  1. In the abstract, the abbreviations (HEMA, CQ, and MEL) should be expanded;

Response: Accordingly, we expanded the abbreviations in the abstract

  1. The keywords should be rearranged after careful consideration;

Response: Accordingly, we rearranged the keywords

  1. In the Result , part 2.2 and part 2.3 have chosen and demonstrated the changes in levels of the stress-responsive genes HMOX-1, NOX4, and oxidative-nitrosative stress parameters iNOS protein, SOD activity after using HEMA and CQ, however, the bioinformatic analysis results in part 2.4 indicate the involvement of oxidative stress in HEMA and CQ-triggered effects. It confuses me, maybe is it proper that this analysis is a hint about the underlying mechanisms of HEMA-and-CQ-induced low cell viability then the oxidative stress is the focus?

Response: We aimed to put the focus not on cytotoxic effects of HEMA and CQ, which has been previously done, but rather on melatonin beneficial effects in HEMA and CQ-treated cells. In order to make this aim clear, we modified the titles of Figure legends. Bioinformatic analyses were done in order to look at the more global picture of biological events involving melatonin in HEMA+CQ-treated hDPCs. This helped in the interpretation of our in vitro  obtained data, by predicting the most likely involved signaling pathways, i.e. altered genes’ expressions. As analyses revealed that the most relevant signaling pathways represent cellular responses to oxidative stress and apoptosis,  we added also in vitro confirmation of melatonin involvement in the regulation of pro/antiapoptotic  pathways.

  1. I think it is still unclear why you choose 5mM HEMA as the treatment.

Response: To evaluate the toxicity of the most common compounds used in restorative dental composites after a 24-h or 48-h exposure period, studies revealed that cytotoxic EC50 values  for HEMA are in the range of  1.77–13.46 mM. Since in the study of Putzeys et al,  concentrations of HEMA found in the uncured conditions were within the range or exceeded the EC50 values, while in the cured conditions, the detected concentrations were lower than the EC50 values, and considering that HEMA has the potential to completely migrate through 300 μm thick dentin disk (A dentin disk of 300 μm represents a situation in which the pulp is near,  but without direct pulp exposure, as in situation of deep carious lesions, which are treated following a complete caries removal technique), we decided for  HEMA concentration of 5mM.

  1. According to Figure1, we can see that MEL1 (0.1 mM) treatment has a positive effect on the amelioration of cell viability in HEMA and/or CQ-supplemented culture, but the cell viability is still lower than in the control group, especially in the MEL1+HEMA+CQ group. Is it possible that a concentration of melatonin lower than 0.1mM have a better effect?

Response: Actually, in our preliminary MTT experiment, conducted on cell culture from one donor, in duplicate, we investigated effects of three melatonin concentrations (0.01mM, 0.1mM and 1mM). However, since at concentration of 0.01mM melatonin did not provide significantly better results compared to 0.1mM (data not shown but are available at request) we decided to go further with experiments with two pharmacological concentrations of 1mM and 0.1mM aiming to highlight differences in concentration-dependent underlying mechanisms of melatonin as well.

  1. The present study has few experiments and seems unsuitable for publication in ijms. The present study only demonstrated that HEMA and/or CQ reduced cell viability and that the addition of melatonin reversed the reduction in cell viability by HEMA and/or CQ, which was accompanied by changes in HMOX1 and NOX4 genes expression and SOD activity and iNOS protein level. However, no experiments have been able to demonstrate that melatonin protection of cell viability from HEMA and/or CQ is mediated by the aforementioned genes or SOD viability.

Response: We did not link directly decreased cell viability by HEMA and CQ  with mechanisms of oxidative stress and apoptosis. In fact, in the text we say that decreased cell viability is accompanied by HEMA and CQ-induced oxidative imbalance. However, our focus was not on HEMA and CQ-cytotoxicity but rather on protective melatonin effects on HEMA and CQ-treated cells, and mechanisms which may underlies it. Both, in vitro and in silico analyses point at melatonin being important in oxidative-nitrosative stress and apoptosis regulation and now  we performed additional experiments on melatonin effects on BAX, BCL-2 and caspase-3 gene expression (which we identified as target genes in bioinformatic analysis) in HEMA and CQ-treated hDPCs. The obtained data  confirmed that HEMA and CQ mediate pro-apoptotic while melatonin acts anti-apoptotic and is able to correct HEMA and CQ-effects. New figure (Figure 5) is included and results are now discussed.

  1. Only one phenotype, cell viability, was studied in the present study, while in a 2015 study, the effects of CQ on cytotoxicity, cell cycle regulation and inflammation-related genes and proteins expression of dental pulp cells has been studied.

Response: This is the first study examining cytotoxicity deriving from concomitant use of HEMA and CQ, which is of importance from the clinical point of view. The aim of this study was to evaluate the beneficial effects and underlying mechanisms of melatonin on cytotoxicity induced by HEMA and CQ in hDPCs, which may contribute to a cost-effective solution (and not just detection) of the HEMA and CQ-related pulp-toxicity problem.  

Reviewer 2 Report

In this study authors evaluate the protective effect of melanin upon HEMA and CQ exposure. The study is interesting, adequately performed and quite well written. It would be interesting if authors could provide a mechanism through which melatonin effects occur and/or provide an in vivo parallelism to the results obtained in vitro.

Author Response

In this study authors evaluate the protective effect of melanin upon HEMA and CQ exposure. The study is interesting, adequately performed and quite well written. It would be interesting if authors could provide a mechanism through which melatonin effects occur and/or provide an in vivo parallelism to the results obtained in vitro.

Response: Thank you for the remarks. Accordingly, we added new experimental data to confirm results from the in silico study regarding melatonin antiapoptotic effects in the hDPCs treated with HEMA and CQ (Figure 5), and discussed them. New paragraph at the end of Discussion was added to emphasize the clinical relevance within the limitations of the present study.

Reviewer 3 Report

In this article entitled “Melatonin mitigates stress effects of HEMA and/or camphor quinone in human dental pulp cells via predicted apoptotic pathways”, the authors aimed to investigate the effect of melatonin on oxidative stress generation induced by HEMA and/or CQ. The results could provide some contribution to the increase in our knowledge on the cytotoxic effects and their mitigation of composite resin components, andcould be considered of interest for the readers. However before it could be considered valid for publication, several points need clarifications and possibly corrections as follows:

INTRODUCTION

Overall well structured, it provides most of the information necessary to understand the scientific background, the knowledge gap and the objectives of the study. However, to facilitate readers' understanding, this reviewer would recommend adding more information on the stress-related molecules examined and the bioinformatic analysis applied.

MATERIAL AND METHODS

The scientific methodology used is in general described in a clear and exhaustive manner. However, this reviewer has a significant concern on the statistical analysis used to evaluate the results. The authors used Mann-Whitney U-test for multiple comparisons, which is inadequate and may cause type I errors. Thus, the authors are requested to redo statistical analysis by using appropriate tests and discuss their findings according to the new results. In addition, giving means and SDs is inappropriate for the data that were analyzed with a non-parametric test.

RESULTS

Figure 1 legends mention "** p<0.01, *** p<0.001 and **** p<0.0001 chemical vs control; # p<0.05, ## p<0.01 78 and ### p<0.001....", although multiple asterisks and sharps do not appear in the figure.

Figure 2: Bars corresponding to the "MEL1 + CQ" group are not visible.

DISCUSSION

The discussion of the results is on the whole well articulated. However, clinical relevance of the results and limitations of the study should be emphasized more.

CONCLUSIONS

The last sentence of this section is merely an inference and I would suggest moving it to the Discussion section.

Minor points

Reference 3 in the first paragraph is not adequate as this does not describe that "eluates can reach dental pulp in sufficiently high concentrations....".

There are a number of words with unnecessary hyphenization throughout the manuscript, which should be corrected.

Author Response

INTRODUCTION

Overall well structured, it provides most of the information necessary to understand the scientific background, the knowledge gap and the objectives of the study. However, to facilitate readers' understanding, this reviewer would recommend adding more information on the stress-related molecules examined and the bioinformatic analysis applied.

Response: Accordingly, we added sentences in the Introduction

MATERIAL AND METHODS

The scientific methodology used is in general described in a clear and exhaustive manner. However, this reviewer has a significant concern on the statistical analysis used to evaluate the results. The authors used Mann-Whitney U-test for multiple comparisons, which is inadequate and may cause type I errors. Thus, the authors are requested to redo statistical analysis by using appropriate tests and discuss their findings according to the new results. In addition, giving means and SDs is inappropriate for the data that were analyzed with a non-parametric test.

Response. Thank you for the remark. You pointed out correctly that Mann-Whitney is not appropriate for multiple comparisons. However, we did not do multiple comparisons but only comparisons between two groups: chemical vs control ( to investigate effects of HEMA or CQ on cells ) and other, unrelated comparison between two groups: chemical with melatonin vs without melatonin (to investigate melatonin effect), but we represented them all on one figure for reasons of technical simplification.

RESULTS

Figure 1 legends mention "** p<0.01, *** p<0.001 and **** p<0.0001 chemical vs control; # p<0.05, ## p<0.01 78 and ### p<0.001....", although multiple asterisks and sharps do not appear in the figure.

Figure 2: Bars corresponding to the "MEL1 + CQ" group are not visible.

Response: Thank you for the remarks. We deleted multiple asterisk symbols from Figure legend and presented the MEL1+CQ group in another, more visible color.

DISCUSSION

The discussion of the results is on the whole well articulated. However, clinical relevance of the results and limitations of the study should be emphasized more.

Response:  Accordingly, we added a paragraph at the end of Discussion 

to emphasize the clinical relevance within the limitations of the present study.

CONCLUSIONS

The last sentence of this section is merely an inference and I would suggest moving it to the Discussion section.

Response: Accordingly, we moved the sentence to the Discussion section.

Minor points

Reference 3 in the first paragraph is not adequate as this does not describe that "eluates can reach dental pulp in sufficiently high concentrations....".

There are a number of words with unnecessary hyphenization throughout the manuscript, which should be corrected.

Responses: We made all the corrections as suggested.

Reviewer 4 Report

The authors aimed to investigate effects and underlying mechanisms of concomitant use of HEMA and CQ with or without co-added melatonin in human dental pulp cells (hDPCs). The authors found that melatonin significantly reduced HEMA and/or CQ-induced up-regulation of HMOX-1 and iNOS expression and SOD activity. Enrichment analyses showed these genes/proteins shared subcellular compartments consisting of apoptotic regulators and are involved in oxidative stress and cell death.

Following are some concerns which need to be addressed by the authors:

1.     To avoid the effects or mechanisms are depend on specific cells, some experiments should be carried out at least two hDPCs.

2.     In Figure 2, the author showed that HO-1 mRNA levels increased significantly following HEMA and CQ treatment. The protein levels of HO-1 should be presented to confirm these results.

3.     Since there are several mechanisms are involved in cell apoptosis, it is highly recommend adding some data about the effect of melatonin on HEMA and CQ-induced cell apoptosis related markers and caspase pathway.

4.     The authors should mention on the physiological relevance of the experimental concentrations of melatonin used in the study and whether such concentrations could become clinically achievable.

Author Response

 Following are some concerns which need to be addressed by the authors:

  1. To avoid the effects or mechanisms are depend on specific cells, some experiments should be carried out at least two hDPCs.

Response: The results presented in the manuscript represent the mean ± SD from 4 experiments (cell cultures were generated from two donors, analyses were performed in duplicate)

  1. In Figure 2, the author showed that HO-1 mRNA levels increased significantly following HEMA and CQ treatment. The protein levels of HO-1 should be presented to confirm these results.

Response: We agree with the reviewer that the findings obtained on RNA level should have been strengthened with findings on protein level. Unfortunately, it is impossible for us at this moment to do the western blot or ELISA experiments as we do not have the antibodies and ordering them would take too much time. However, our results showed that HEMA and CQ-mediated up-regulation of HMOX-1  accompanied HEMA and CQ-mediated iNOS upregulation which could suggest another known HMOX-1-mediated signaling pathway since  HMOX-1 is identified as a critical instigator of NO-derived oxidative stress ( https://doi.org/10.1161/CIRCRESAHA.118.312910).

  1. Since there are several mechanisms are involved in cell apoptosis, it is highly recommended adding some data about the effect of melatonin on HEMA and CQ-induced cell apoptosis related markers and caspase pathway.

Response: Accordingly, we performed additional experiment in order to investigate effects of melatonin on BAX, BCL-2 and caspase-3 genes expression (which we identified as significant target pathways in bioinformatic analysis) in HEMA and CQ-treated hDPCs and the new  results are presented  (Figure 5) and discussed.

  1. The authors should mention on the physiological relevance of the experimental concentrations of melatonin used in the study and whether such concentrations could become clinically achievable.

Response: The melatonin concentrations used in the present experiments  were pharmacological concentrations and we aimed to prelude forthcoming in vivo investigations of local melatonin application, under composite filling in proper formulation to alleviate potentially harmful effects of HEMA and CQ on pulp cells. New paragraph at the end of Discussion was added to emphasize the clinical relevance within the limitations of the present study.

Round 2

Reviewer 1 Report

The present study has few experiments and does not seem suitable for publication in ijms. More experiments and a more interesting story are needed.

Author Response

Thank you for the remarks. In the light of new findings of higher incidence of postoperative sensitivity in type 2 diabetic patients as well as investigation of dental pulp of diabetics, we have rewritten the manuscript according to new proposed hypothesis. Two new figures and new table have been added. The title is modified accordingly.

Reviewer 2 Report

Authors have responded to my requests.

Author Response

Thank you for the suggestions that improved the manuscript

Reviewer 3 Report

In this revision, the authors gave a careful reply to my comments and revised the manuscript according to them. I think these modifications improved the manuscript.

Author Response

Thank you for the suggestions that significantly improved our manuscript.

Reviewer 4 Report

There are not any substantive modifications, compared with the previous version. I can't approve this manuscript to be published in this journal.

Author Response

Thank you for the remarks. In the light of new findings of higher incidence of postoperative sensitivity in type 2 diabetic patients as well as investigation of dental pulps of diabetics, we have rewritten the manuscript according to new proposed hypothesis. Two new figures and new table have been added. The title is also modifed accordingly.

Round 3

Reviewer 1 Report

1.  The format of “A/B/C” or  “A/B/C” or  “(A)/(B)/(C)” in each figure should be unified;
2. In the "Introduction", "hDCPs" should be corrected;
3. In the end of Figure 4 legend, are they ("(5 mM); CQ- camphorquinone (1mM)")  duplicated words?
4. The methods for how to measure the protein expression of Melatonin in the dental pulp should be added

Reviewer 4 Report

I recommend to publish in this present form